# Fodor and Pylyshyn's Legacy – Still No Human-like Systematic Compositionality in Neural Networks

## Abstract

Strong meta-learning capabilities for systematic compositionality are emerging as an important skill for navigating the complex and changing tasks of today's world. However, in presenting models for robust adaptation to novel environments, it is important to refrain from making unsupported claims about the performance of meta-learning systems that ultimately do not stand up to scrutiny. While Fodor and Pylyshyn famously posited that neural networks inherently lack this capacity as they are unable to model compositional representations or structure-sensitive operations, and thus are not a viable model of the human mind, Lake and Baroni recently presented meta-learning as a pathway to compositionality. In this position paper, we critically revisit this claim and highlight limitations in the proposed meta-learning framework for compositionality. Our analysis shows that modern neural meta-learning systems can only perform such tasks, if at all, under a very narrow and restricted definition of a meta-learning setup. We therefore claim that 'Fodor and Pylyshyn's legacy' persists, and to date, there is no human-like systematic compositionality learned in neural networks.

## 1 Introduction

Meta-learning, or *learning to learn* from different situations, is an interesting challenge closely related to human intelligence. It is a core element of our educational system that we learn *how to learn* without explicit prior knowledge about each situation in life, as their variations are manifold. Similarly, the use of language embodies this adaptability, requiring the integration of learned rules with contextual nuances to navigate both familiar and novel scenarios. Language exemplifies how humans apply systematic generalization, seamlessly combining learned grammatical structures and vocabulary to create and interpret new expressions. This dynamic interplay between rules and context bridges the abstract principles of meta-learning with the practical mechanisms that underlie communication and cognitive reasoning.

The principle of compositionality is a key challenge for artificial neural networks, as it requires the ability to develop systematic representations and behaviors. Unlike humans, neural models often struggle to generalize such rules (Nezhurina et al. 2024, Wüst et al. 2024a, Bayat et al. 2025) across contexts, reflecting fundamental gaps in their representational and operational frameworks. Because artificial neural networks are constrained by their reliance on finite representational spaces and distributed encoding schemes, these limitations manifest themselves in their difficulty in applying composition rules consistently across scenarios. While humans can effortlessly recombine learned concepts to interpret novel sentences or solve unique problems, neural networks lack the inherent transparency, flexibility, and reflexivity to perform similar feats. Their opacity, driven by distributed representations, hinders their ability to systematically manipulate components and infer relationships.

Lake and Baroni [2023] introduced a meta-learning framework attempting to mitigate these challenges by introducing episodic training tasks that require rule inference. The framework involves presenting neural networks with support examples governed by hidden grammars and testing their ability to generalize these rules. This episodic approach aims to train networks for systematic generalization, using meta-learning principles to approximate human-like reasoning. They claimed to overcome some fundamental limitations of neural networks, prominently stated by Fodor and Pylyshyn [1988]. However, there is also plenty of evidence of the limitations of modern deep learning models with human-like capabilities in language understanding that rely on systematic compositional reasoning (Deletang et al. 2023, Zhang et al. 2023, Dziri et al. 2024, Mészáros et al. 2024, Bayat et al. 2025), and we provide further insights that even Lake and Baroni's model fails to prove its systematic behavior in several instances.

Despite its potential, the framework's reliance on learned distributions and predefined rule shapes limits its scope. Generalization remains limited to permutations of known rules rather than the discovery of entirely new principles. The difficulty of scaling to complex tasks with deeper nesting underscores the persistent gaps in achieving true human compositional reasoning. Lake and Baroni's framework provides valuable insights, but also highlights the need for innovation in neural network training and evaluation to overcome these limitations, as behavioral similarities may mask fundamental differences in underlying mechanisms.

Thus, in this paper we argue that: **Neural networks have not yet achieved learning systematic compositional abilities**. Specifically, based on a case of effective criticism of Lake and Baroni's framework, we outline how to argue, test, and train for systematic generalization and compositionality, and demonstrate the relevance of our position (*c.f.* Figure 1 for a schematic).

We develop this position as follows: **(I)** We identify Fodor and Pylyshyn's main arguments in the context of the compositionality challenge for artificial neural networks and locate the nature of Lake and Baroni's approach in refuting Fodor and Pylyshyn's claims that neural networks cannot reliably develop *compositional representations* and *structure-sensitive operations*. **(II)** We show that within their setup, the model exhibits various *non-systematic* behaviors that can not be considered human-like and clearly violates structure-sensitivity. **(III)** We argue for several necessary aspects of training and evaluation of meta-learning systems to achieve and assess their systematicity, taking into account the relevance of compositional representations and structure-sensitive operations. **(IV)** We adapt the arguments of Fodor and Pylyshyn in light of the modern development of deep learning systems to argue for a future of models capable of learning symbolic representations for artificial cognition and representation learning.

## 2 The Challenge of Compositionality

### 2.1 Fodor and Pylyshyn's legacy

In their influential 1988 paper, *Connectionism and cognitive architecture: A critical analysis*[1], Fodor and Pylyshyn claim that artificial neural models are unsuitable for modeling the human mind on a cognitive level. They review several arguments for the combinatorial structure of mental representations, highlighting the systematicity of these representations that follow the compositional nature of cognitive capabilities; the ability to understand some given thoughts implies the ability to understand various thoughts not only with semantically related content but also of a more complex combinatorial structure. Nevertheless, they also consider the possibility that artificial neural networks may play a role in *implementing* cognition.

**Differentiating neural networks and symbolic systems.** They begin by discussing the disagreement about the nature of mental processes and mental representations between the so-called Connectionist approach, which focuses on artificial neural networks, and the Classical approach, which favors symbolic systems like Turing machines for modeling cognitive abilities. They emphasize that it is neither about the explicitness of rules, nor about the reality of representational states, nor about non-representational architecture, since a "Connectionist neural network can perfectly well implement a Classical architecture at the cognitive level"[2]. While both "assign semantic content to *something*"[3],

---

[1][Fodor and Pylyshyn, 1988].
[2]Ibid., p.11.
[3]Ibid., p.12, emphasis in original.

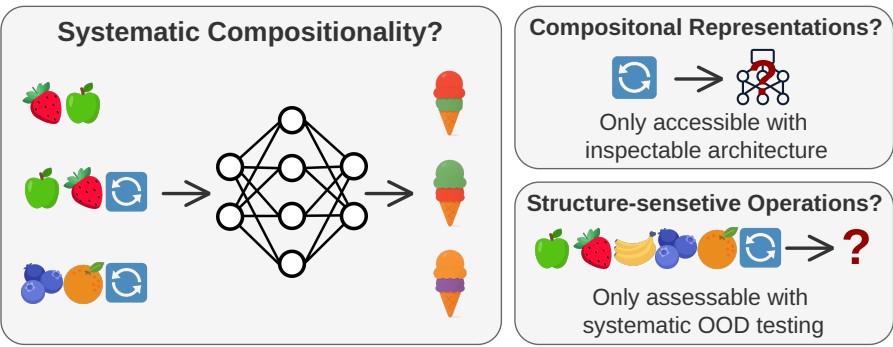

Figure 1: **The challenge of claiming and testing systematic compositionality.** Given the undisputed importance of *compositional representations* and *structure-sensitive operations* for systematic compositionality, their evaluation remains crucial and challenging. While structure-sensitivity can be assessed by comprehensive OOD testing, the investigation of representations requires some inspectable model architecture.

it is identified as the central difference that they disagree about what primitive relations hold between these content-bearing entities. The sole importance of causal connectedness in neural networks is contrasted with a range of semantic and structural relations in symbolic systems. Only the sensitivity to both semantic and structural relations is expected to allow a commitment to the compositionality of mental representations with combinatorial syntax and semantics. Furthermore, the operations that models perform in transforming one representation into another are sensitive to the structure of these representations and not only to their semantics.

**Productivity, compositionality, and systematicity of cognitive ability.** The need for these two properties of symbol systems, *compositional representations* and *structure-sensitive operations*, is justified by "three closely related features of cognition: its productivity, its compositionality, and its inferential coherence"[4]. Only structure-sensitive operations combined with a combinatorial structure and semantics of representations can account for the (under appropriate idealization) *unbounded capacities* of a representational system. Similarly, cognitive capacities are systematic in that the ability to produce or process some representations is syntactically linked to the ability to produce or process certain other representations without relying on the processing of any particular semantics, e.g., understanding the form of the expression $(A \land B) \to A$ implies the ability to understand the expression for any substituents of $A$ or $B$. In fact, systematicity makes a stronger argument by using a weaker assumption, since "[p]roductivity arguments infer the internal structure of mental representations from the presumed fact that no one has *finite* intellectual competence [and by] contrast, systematicity arguments infer the internal structure of mental representations from the patent fact that no one has *punctuate* competence." [5] Closely related to systematicity is the compositionality of mental representations, since representational abilities can be linked not only syntactically but also semantically. It is important to note here that not every mental representation is expected to be compositional, e.g., the understanding of some expressions in natural language, since "similarity of constituent structure accounts for semantic relatedness between systematically related sentences only to the extent that the semantic properties of the shared constituents are context independent."[6] A final cognitive feature is the systematicity of inference. Recalling the example of the conjunction $A \land B$ entailing its constituent $A$, it is not only the mental representation of the understanding of this rule that is systematic, but also its application for coherent inference between thoughts, which in turn requires the structure-sensitivity of operations in symbolic systems.

**Neural networks for implementing symbol systems.** Finally, Fodor and Pylyshyn comment on treating Connectionism as an implementation theory for cognitive architecture. They "have no principled objection to this view"[7]. However, they emphasize that if neural networks are only a *method for implementing* cognitive architecture, their internal states are useless for understanding the

---

[4]Ibid., p.33.

[5]Ibid., p.40, emphasis in original.

[6]Ibid., p.42.

[7]Ibid., p.67.

nature of mental representations and therefore "irrelevant for psychological theory"[8]; neural networks would only be neurological, and the need for and relevance of symbol systems for modeling cognition would remain untouched.

## 2.2 Lake and Baroni's objection

**Compositional seq2sec tasks.** Lake and Baroni present their work as evidence against the claims of Fodor and Pylyshyn. They present a meta-learning framework that they claim achieves or exceeds human-level systematic generalization in its evaluations. Their experimental setup is based on sequence-to-sequence (sec2sec) transduction tasks, considering sequences generated over 8 pseudolanguage tokens $u \in U$ for the input domain $X = U^*$, while the output domain $Y = C^*$ comprises sequences generated over 6 different color tokens $c \in C$. Both domains are connected by a transduction grammar, i.e. a set of production rules that define how a sequence of input tokens is translated into a color sequence. Each rule is of two kinds; it can state a primitive transduction rule $u \to c$, which simply maps an input token to an output token; otherwise it states a unary operation $v_1 u \to f_u(v_1)$ or a binary operation $v_1 u v_2 \to g_u(v_1, v_2)$, where each $f$ is some $n$-fold ($n \leq 8$) repetition, each $g$ is some combination of repetition, permutation, and concatenation. Each $v_i$ is either a single token $u_i$ or the entire preceding or succeeding token string $x_i$. By iteratively composing these rules, such a grammar generates a set of translatable input sequences $\bar{X} \subseteq X$.

**Seq2seq meta-learning framework for evaluation of human systematic generalization.** With these transduction tasks, Lake and Baroni set up a meta-learning framework with *episodes* associated with different transduction grammars. Each episode combines a *SUPPORT* set of input-output transduction pairs and a *QUERY* set of input-output pairs, each pair being consistent with the associated grammar. The query outputs are hidden, and the task is to reproduce them with the support examples as the only information given; the underlying transduction grammar also remains hidden. In this way, it is not necessary to infer the grammar rule explicitly. Nevertheless, the ability to implicitly extract or hypothesize the actual grammar rules is expected to be essential for reliably deriving the correct query outputs. A standard seq2seq transformer network is now trained on query examples from different episodes. The transformer encoder processes a query input combined with the support pairs of its episode as context, and the transformer decoder generates an output sequence.

## 3 Systematicity through Meta-Learning

In the following, we illustrate the limitations of current neural network approaches by examining the systematicity achieved by Lake and Baroni's meta-learning approach. After revealing a severe lack of compositionality in their framework, we propose how to better test and train for systematic generalization and compositionality with meta-learning systems. In doing so, we highlight the ongoing challenges associated with compositional representations and structure-sensitive operations.

**Locating Lake and Baroni's approach.** In order to evaluate the proposed framework for systematic generalization by meta-learning neural networks with respect to Fodor and Pylyshyn's claims, we will first clarify which of Fodor and Pylyshyn's arguments Lake and Baroni are referring to, since they primarily present an implementation of what they claim is a human-like systematic capability, but directly address a challenge. They themselves situate their work as a contribution to the line of argument that Fodor and Pylyshyn's statements no longer apply to current model architectures; they are not criticizing the properties of human cognition, but the alleged inability of neural networks to reliably develop *compositional representations* and *structure-sensitive operations*. By focusing on behavioral tests rather than ablation studies that directly examine the structure of learned representations, Lake and Baroni emphasize the *structure sensitivity* and *systematicity* of their model, which is crucial for demonstrating compositional abilities and coherent behavior. Furthermore, they present their meta-learning framework for compositionality to systematically train neural networks with these abilities. While a single neural network with compositional abilities would not contradict Fodor and Pylyshyn, who did not claim any *limits on implementability* of cognitive abilities, a framework that reliably achieves *compositional abilities by stochastic learning* methods would actually contradict their main point of criticism. Unfortunately, we will see in the following section that the model trained on meta-learning still fails to reliably demonstrate compositional ability in several examples.

---

[8]Ibid., p.65.

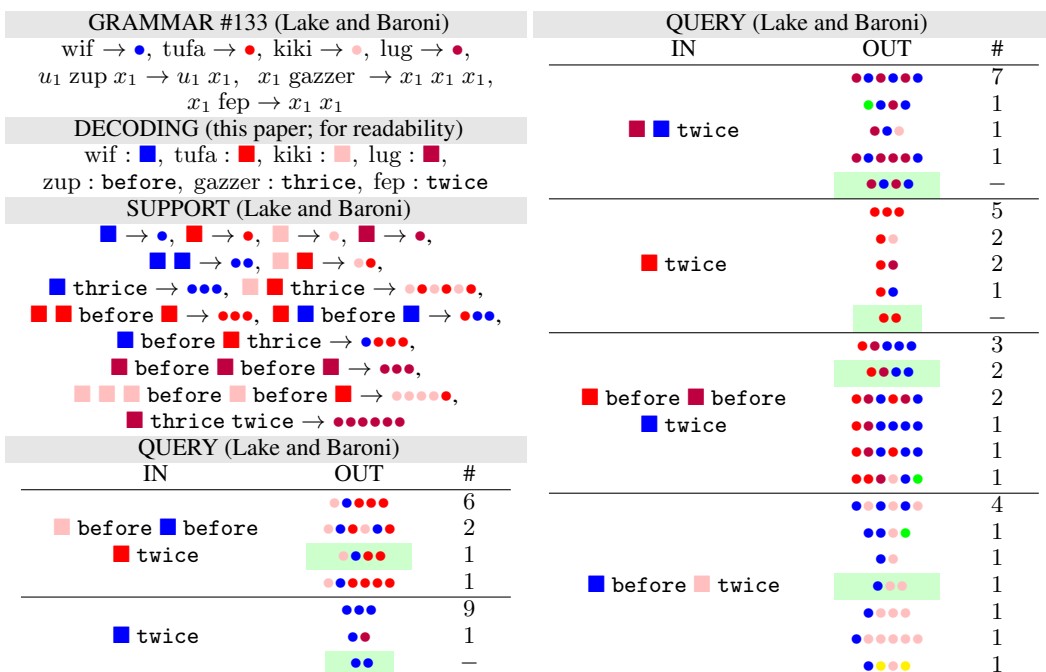

Table 1: Episode #133 with 10 evaluations for each query example; SUPPORT and QUERY are decoded for better readability. Expected outputs backed with green. The model shows *incoherent processing* and *systematically mistakes* `twice` for `thrice`. Further results can be found in Appendix A.2.1. (Best viewed in color.)

### 3.1 Examining the lack of compositionality

Lake and Baroni mention that generalization beyond training occurs only with respect to *new combinations* of three grammar rules from the same set of grammar rules used during training. However, if we consider the invariance under the atomic assignments of colors to language tokens and the mere labeling of operations, we find that 179/200 validation episodes have a combination of non-primitive grammar operations that were already present in the 100000 training episodes. (See Appendix A.1 for details.) Thus, even if the model achieves highly systematic results on the test episodes, this could be due to memorization of the experienced operation patterns and learning to extract the correct labels from the episode's support examples. However, we can even show that there is non-systematic behavior within their repository of test episodes; we re-evaluate their pre-trained $'$net $-$ BIML $-$ top$'$ model on the same set of $'$algebraic$'$ test episodes, with the only difference that we did 10 evaluations of all query examples for each test episode, for statistical purposes, similar to the one episode they further evaluated against human performance. We find that the model performs worst on episodes #133, #32, and #122, with accuracies of only 41%, 52%, and 54% on the query examples, respectively. (See the next paragraph and the Appendix A.2 for details.)

**Failure in rule extraction.** Further investigation of Episode #133 (see Table 1) reveals that the model has trouble correctly processing the semantics of the language token ⟨fep⟩ with the hidden grammar rule $x_1$ fep $\rightarrow x_1\ x_1$ and will therefore call it ⟨twice⟩ and confuse it up with the token ⟨gazzer⟩ (with $x_1$ gazzer $\rightarrow x_1\ x_1\ x_1$) which we will call ⟨thrice⟩. It seems to have a problem with the only example with ⟨twice⟩, ⟨■ thrice twice $\rightarrow$ ●●●●●●⟩, which also happens to contain ⟨thrice⟩. But since ⟨thrice⟩ has several iconic examples in the support, it is expected that a reasoner with compositional skills will be able to systematically use a single example and remain consistent with the rest of the support information. Considering the examples ⟨■ $\rightarrow$ ●⟩, ⟨■ $\rightarrow$ ●⟩, ⟨■ thrice $\rightarrow$ ●●●⟩, human systematicity would at least suspect some semantics of ⟨twice⟩ that are different from those of ⟨thrice⟩.

**Non-systematic parsing.** Interestingly, the hidden grammar allows for an ambiguous interpretation of nested transduction queries, which would normally be a challenge for a systematic reasoner. For example, the query ⟨■ before ■ twice⟩ could be parsed as either ⟨■ before (■ twice)⟩ (marked

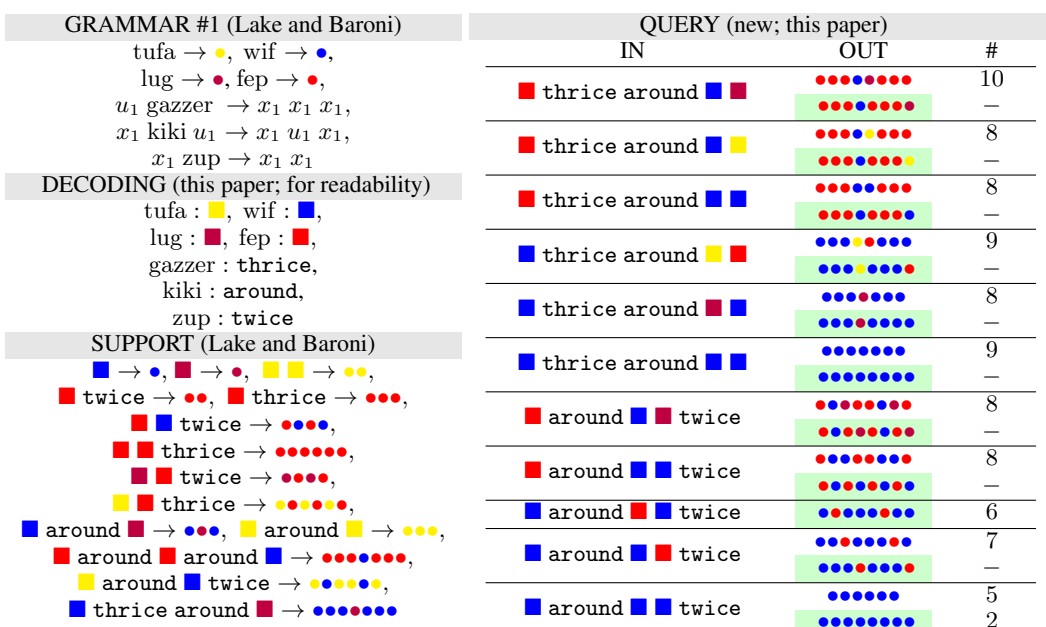

Table 2: Episode #1 with our own query examples and with 10 evaluations for each input; SUPPORT and QUERY are decoded for better readability. Expected outputs backed with green. Further results can be found in Appendix A.5 (Best viewed in color.)

as the target by Lake and Baroni ) or ⟨(■ before ■) twice⟩, and similarly for a query with ⟨thrice⟩. But the support example ⟨■ before ■ thrice → ●●●●⟩ should at least induce a bias toward the intended processing. But the answers to this challenge also lack systematicity; while the common mistakes ⟨■ before ■ before ■ twice → ●●●●●⟩ and ⟨■ before ■ before ■ twice → ●●●●●⟩ could be explained by processing ⟨$u_1$ before ($u_2$ before ($u_3$ thrice))⟩ while, in contrast, a similar explanation to the the error ⟨■ before ■ twice → ●●●●●⟩ would be the parsing ⟨($u_1$ before $u_2$) thrice⟩. We will further discuss the importance of systematicity for meta-learning systems in Section 3.2.

**Violating structure-sensitivity.** Besides both previous failure modes that are related to incompetence in extracting information from the support examples, we also found query examples for episode #1 that reveal additional non-systematicity (see Table 2 or Appendix A.2 for extended version). For queries with the patterns ⟨$u_1$ thrice around $u_2$ $u_3$⟩ and ⟨$u_1$ around $u_2$ $u_3$ twice⟩ we first see that the model never parses ⟨around⟩ as intended. Instead of ⟨(($u_1$ thrice) around $u_2$) $u_3$⟩ and ((⟨$u_1$ around $u_2$⟩) $u_3$) twice⟩, the stable output can be explained by parsing ⟨around⟩ as intended. Instead of ⟨($u_1$ thrice) around ($u_2$ $u_3$)⟩ and (($u_1$ around ($u_2$ $u_3$)) twice⟩ – except for the cases, ⟨■ thrice around ■ ■⟩, ⟨■ thrice around ■ ■⟩, ⟨■ around ■ ■ twice⟩, *where it would make no difference*! Only the (also ambiguous) case ⟨■ around ■ ■ twice⟩ is processed correctly in 6/10 cases – but with even worse performance than with the unambiguous examples. Despite the structural similarity to the other query examples, down to the color combination, we see a non-systematic deviation in the response, which raises doubts about compositional skills.

**Limits in productivity.** Finally, we would like to point out that Lake and Baroni's setup only allows the model to process input sequences of up to 10 tokens and generate output sequences of up to 8 color tokens (which further restricts the allowed input sequences). This limits the ability to test more complex input sequences and thus to assess the *productivity* of the model's ability.

## 3.2 Our position on meta-learning systems

We now discuss whether meta-learning, beyond Baroni's framework, could be a promising approach towards human-like compositional skills, despite the demonstrated limitations in the specific setup. Meta-learning systems aim to emulate human-like learning by incorporating systematicity and flexibility into their architectures. These systems aim to (1) generalize beyond training examples by

inferring composition rules from limited examples, (2) adapt to novel contexts with flexibility as a key expectation, allowing systems to quickly transfer skills to new domains with minimal retraining, and (3) mirror human-like cognition by ensuring that error patterns and reasoning paths are still systematic, explainable, or even self-correcting.

**Weakness of non-reflective training.** A major shortcoming of Lake and Baroni's work is the use of a one-shot prediction approach. Models are trained to perform a direct transduction on the presented support examples without any intermediate reflection or validation steps. To ensure the systematic production of results, we argue that it is of primary importance for meta-learning models to iteratively extract, validate, and correct their current beliefs in the extracted rules. In the previous section, we showed that Lake and Baroni's models fail to validate extracted rules against the support, and thus systematically fail to correctly extract (and consequently apply), for example, the `twice` rule.

**Focus on systematicity rather than productivity.** Given the role that underlying grammars play with respect to meta-learning, or more precisely, non-meta-learning problems, any of today's modern transformer systems can be broken by feeding them increasingly complex problems until the models are no longer expressive enough to capture the problem as a whole. This can be due to the depth of rule nesting or simply the length of the input. While the general ability to learn to transcribe rules is certainly a prerequisite for meta-learning systems in the particular setting discussed, one would not necessarily deny such systems the ability to perform meta-learning reasoning even if they fail at such tasks for the reasons discussed above. When discussing meta-learning tasks, the focus is not on the ability to derive rules of arbitrary complexity – which is a problem of classical machine learning – but on the ability of these models to systematically discover, verify, apply, and combine these rules, or to systematically learn from their mistakes. Compared to human reasoning [Nezhurina et al., 2024, Wüst et al., 2024b], meta-reasoning abilities are not judged by the ability to produce transductions in a one-shot fashion, but rather with a focus on the correctness of the result in the *final output*. Therefore, we make the following claim:

*Claim* 1. A characteristic of ***successful* meta-learning systems** is the ability to consistently **abstain from *non-systematic* errors**.

Our primary concern is with the consistency of model behavior, and we therefore distinguish between systematic and non-systematic errors. Systematic errors can result from incorrect assumptions inherent in the model, which are then applied systematically. In our setting, this may involve assumptions about the unique interpretation of rules – see, e.g., our discussion of potentially ambiguous rule interpretations in Lake and Baroni – and, more generally, may be due to exogenous factors and implicit assumptions not captured during the training phase. While such errors may not produce the desired result, they follow a systematicity that suggests that the model might have been able to learn the correct rules given the correct underlying assumptions. The lack of systematicity, however, is a much larger error. Here, models may exhibit erratic 'glitches' that result in non-human-like behavior that lacks any systematicity. Since the underlying reasons for such behavior may not be generally understood, it is unclear how to handle and correct such errors. Finally, we derive two positions regarding essential aspects of evaluation and training of successful meta-learning systems:

**Position on evaluation.** Assessing and postulating systematic or compositional skills in neural networks requires either the direct evaluation of the model's internal representations, which would require an inspectable or explainable network architecture, or the use of comprehensive ablation studies that systematically test a model's behavior in out-of-distribution situations.

**Position on implementation and training.** To achieve compositionality and systematicity within the discussed meta-learning tasks, the presence of symbolic representations within neural networks is essential to ensure consistent application and composition of rules. We would like to emphasize that while Fodor and Pylyshyn remain unrefuted in the general analysis, today's discussion of modern neural network architectures is constantly evolving to develop symbolic representations, e.g., in the form of circuits [Olah et al., 2020, Wang et al., 2022, Conmy et al., 2023, Hanna et al., 2024]. These explicit representations are important building blocks that promote consistent behavior and allow explicit reflection and iterative correction of possible inconsistencies in the extracted rule sets. Finally, it is important to note that reflective behavior is not likely to evolve from training on one-shot transduction tasks, but *requires models to have the ability to iterate, validate, and correct over the extracted rule sets*. Recently, important breakthroughs in this direction have been made in RL training of language reasoning models [Stiennon et al., 2020, Ouyang et al., 2022, Bai et al., 2022, Lee et al., 2023, DeepSeek-AI et al., 2025].

## 4 Related Work

**Human-like compositionality.** Regarding the importance of compositionality for cognitive abilities, Fodor and Lepore [2001] and Fodor [2001] extend the discussion of Fodor and Pylyshyn [1988] on the compositional nature of language and thought. While (natural) language contains some non-compositional structures due to *context sensitivity*, compositionality is argued to be essential for (a language of) thought. This is in line with recent work by Fedorenko et al. [2024], which tries to find evidence that language is primarily a tool for communication rather than for thinking.

**Compositionality in neural networks.** Besides Lake and Baroni [2023], there is older as well as recent work trying to demonstrate compositional or meta-learning capabilities achieved with neural network architecture [Botvinick and Plaut, 2004, Santoro et al., 2016, Park et al., 2024, DeepSeek-AI et al., 2025]. Other work is proposing frameworks for learning and assessing compositional skills [Petrache and Trivedi, 2024, Sinha et al., 2024] or other intelligent behavior [Chollet, 2019] and Bayat et al. [2025] is introducing memorization-aware training to tackle overfitting to spurious correlations encountered in training.

**Limitations in systematicity.** Several works evaluate and demonstrate the limitations of modern AI models in compositional or systematic generalization tasks [Bender et al., 2021, Deletang et al., 2023, Dziri et al., 2024, Mészáros et al., 2024, Nezhurina et al., 2024, Zhang et al., 2024, Wüst et al., 2024b] and there is a direct response to the work of Lake and Baroni, which presents problems of non-systematic behavior [Goodale and Mascarenhas, 2023].

**Importance of symbolics.** There is also more recent work that emphasizes the importance of symbolics. Ellis et al. [2020] presents a machine learning system that uses neurally guided program synthesis to learn to solve problems. Wüst et al. [2024a] further demonstrates the advantages of using program synthesis for unsupervised learning of complex, relational concepts from images, focusing on the benefits in terms of generalization, interpretability, and revisability. Stammer et al. [2024b], on the other hand, investigated the benefits of symbolic representations for improving the generalization and interpretability of low-level visual concepts. The position of the importance of symbols for AI explanations is further discussed by Kambhampati et al. [2022]. The approach of Dinu et al. [2024] combines generative models and solvers by using large language models as semantic parsers. Shindo et al. [2025] models the human ability to combine symbolic reasoning with intuitive reactions by a neuro-symbolic reinforcement learning framework.

## 5 Alternative Views

Historically, [Fodor and Pylyshyn, 1988] argued for the emergence or implementation of symbolic reasoning structures within neural networks as a necessary aspect of achieving human-like meta-learning. However, the meta-learning considerations discussed in their paper and ours focus strongly on the learning of logical and arithmetic rules, where concepts can be reduced to symbolic representations. These representations, therefore, naturally fit well with the capabilities of symbolic reasoners but leave out other possible forms of meta-learning systems. The consideration of different modalities, e.g., for the composition of visual patterns or motion sequences, can be a strong hurdle for classical symbolic systems. Such domains, which do not operate on discrete 'crystallized' symbols but rather on abstract 'fluid' concepts, still lack a well-defined notion of what constitutes meta-learning within them. As a consequence, it is unclear how to measure and systematically evaluate the meta-learning abilities of models in possible benchmarks.

**Untargeted emergence of systematic reasoning.** Even without training towards meta-learning models, LLMs exhibit some emergent abilities for various tasks [Brown et al., 2020, Wei et al., 2022a, Schaeffer et al., 2024]. While 'true' understanding of the world might only be achieved via (embodied) interaction [Lipson and Pollack, 2000, Gupta et al., 2021, Zečević et al., 2023], some works have argued that such abilities might even be learned through mere passive observation [Lampinen et al., 2024], while other approaches argue for the value of self-explanatory guided learning [Stammer et al., 2024a]. Considering the underlying aspect of systematic learning and reasoning, several works have been able to distill symbolically acting *circuits* that emerge during training from LLMs [Olah et al., 2020, Wang et al., 2022, Conmy et al., 2023, Hanna et al., 2024]. In light of these results, it remains to be seen whether meta-learning abilities of language reasoning models might also emerge as a consequence of pure scaling laws [Sutton, 2019, Kaplan et al., 2020, Bubeck et al., 2023].

# 6 Position Summary and Discussion

For this final section, we will reiterate the key points that make up our position (see Sec. 1) and that we believe are important aspects of the goal of achieving meta-learning models capable of human-like systematic compositionality: **(I) Criteria for compositionality.** The main criteria for models with productive, systematic, and compositional capabilities remain *compositional representation* and *structure-sensible operations*. **(II) Non-systematic behavior.** Since Lake and Baroni's model exhibits various non-systematic behaviors, it fails to demonstrate human-like compositional learning capabilities, and further refutes the presented claims that their meta-learning framework achieves human-like systematic generalization. **(III) Assessment of compositionality.** Systematic testing of multiple types of out-of-distribution episodes is necessary to assess compositional abilities and structure-sensitive operations. **(IV) Emergence and Learning of Symbolic Representations.** Meta-learning systems need to support the emergence of compositional symbolic representations during training. For this, we expect training tasks and model architectures that make iteration, self-validation, and self-correction over the extracted rule sets possible and necessary. The limitations of current neural models underscore the importance of hybrid architectures that integrate the strengths of symbolic and connectionist paradigms. Key advances in this direction include systematicity, reflective reasoning, and scalability.

**Systematicity.** Embedding mechanisms for representing and manipulating composition rules in neural architectures is a key step toward improving generalization. In this paper, we argue that a central property of meta-learning systems is the ability to refrain from non-systematic errors. This includes the ability to represent explicit rules and apply them consistently across different contexts. Models that incorporate such structure are better able to generalize compositionally and avoid brittle behavior when encountering novel combinations of inputs.

**Reflection.** Embedding iterative, self-correcting processes into models is essential for emulating human-like adaptability. A distinctive ability of human reasoning is to reflect on and refine a set of currently hypothesized rules. Extracting and validating rules from support examples can become increasingly complex as the number of examples grows, often scaling exponentially. While one-shot models can perform well within limited problem sizes, they are ultimately constrained by fixed model capacity. We therefore argue for reflective learners –models capable of iteratively refining and self– correcting their internal representations. This approach enables repeated validation of inferred rules and aligns with recent successes in general language reasoning through iterative prompting and reasoning [Wei et al., 2022b, Yao et al., 2024, DeepSeek-AI et al., 2025]. Unlike one-shot answers, this iterative behavior supports progressive improvement and robust generalization.

**Scalability, memory and context.** Enabling models to dynamically extend rule sets and adapt to new tasks is critical to mirroring human flexibility. A core requirement for reflective reasoning is the ability to store and manipulate representations of a model's current beliefs. This includes mechanisms for reading and updating the memory as new information becomes available. When applying rules to a query, a model may also need to track contextual factors-such as the nesting depth of current rules-which requires memory components that can generalize beyond a fixed number of parameters. Thus, overcoming the limitations of static architectures requires models that can manage dynamic memory and evolving contexts to support scalable reasoning across diverse and complex tasks.

# 7 Conclusion

While the importance of *compositional representations* and *structure-sensitive operations* for human-like systematicity remains, the previous consideration allows the training and testing of artificial neural networks that encourage the development of such properties. By bridging the gap between symbolic and connectionist principles, hybrid architectures may be particularly promising, since they do not suffer from the limitations of neural networks without symbolic machinery as specified by Fodor and Pylyshyn. Overall, the continued relevance of Fodor and Pylyshyn's critique underscores the challenges of developing systems capable of systematic generalization and compositional reasoning. While meta-learning frameworks represent significant progress, they do not address fundamental limitations. Future advances must embrace integrative approaches that combine the strengths of symbolic and connectionist paradigms, paving the way for a more robust understanding of artificial cognition. By addressing these challenges, we can move closer to realizing the vision of human-like artificial intelligence.

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

# A APPENDIX: Fodor and Pylyshyn's Legacy – Still No Human-like Systematic Compositionality in Neural Networks

## A.1 Note on analysis for different combinations of non-primitive grammar operations

In section 3.1 we state that only 179/200 validation episodes have a combination of non-primitive grammar operations that were already present in the 100000 training episodes. This is the result of counting every different combination of 3 operations, unary ($v_1u \rightarrow f_u(v_1)$) or binary ($v_1uv_2 \rightarrow g_u(v_1, v_2)$), present in both training and validation episodes, when abstracting the individual function names $u$.

## A.2 Extended outputs for Lake and Baroni's meta-learning testing episodes

Below we include the full set of grammar rules, support examples, and query examples from Lake and Baroni's meta-learning. We re-evaluated their pre-trained $'net - BIML - top'$ model on the same set of $'algebraic'$ test episodes. Here we report the results for #133, #132, #122, and the modified #1.

### A.2.1 Complete responses for Lake and Baroni's meta-learning testing-episodes #133.

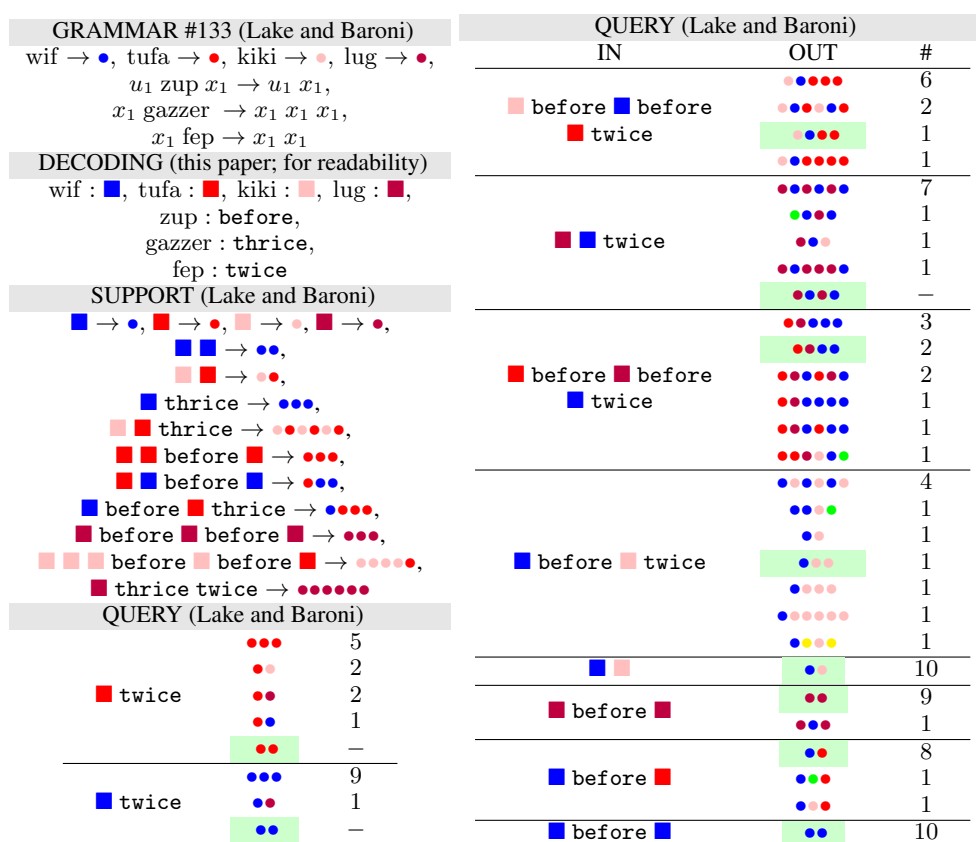

Table 3: Episode #133 with 10 evaluations for each query example; decoded for better readability. Expected outputs backed with green. The model shows *incoherent processing* and *systematically mistakes* `twice` for `thrice`. (Best viewed in color.)

582 **A.3 Complete responses for Lake and Baroni's meta-learning testing-episode #32.**

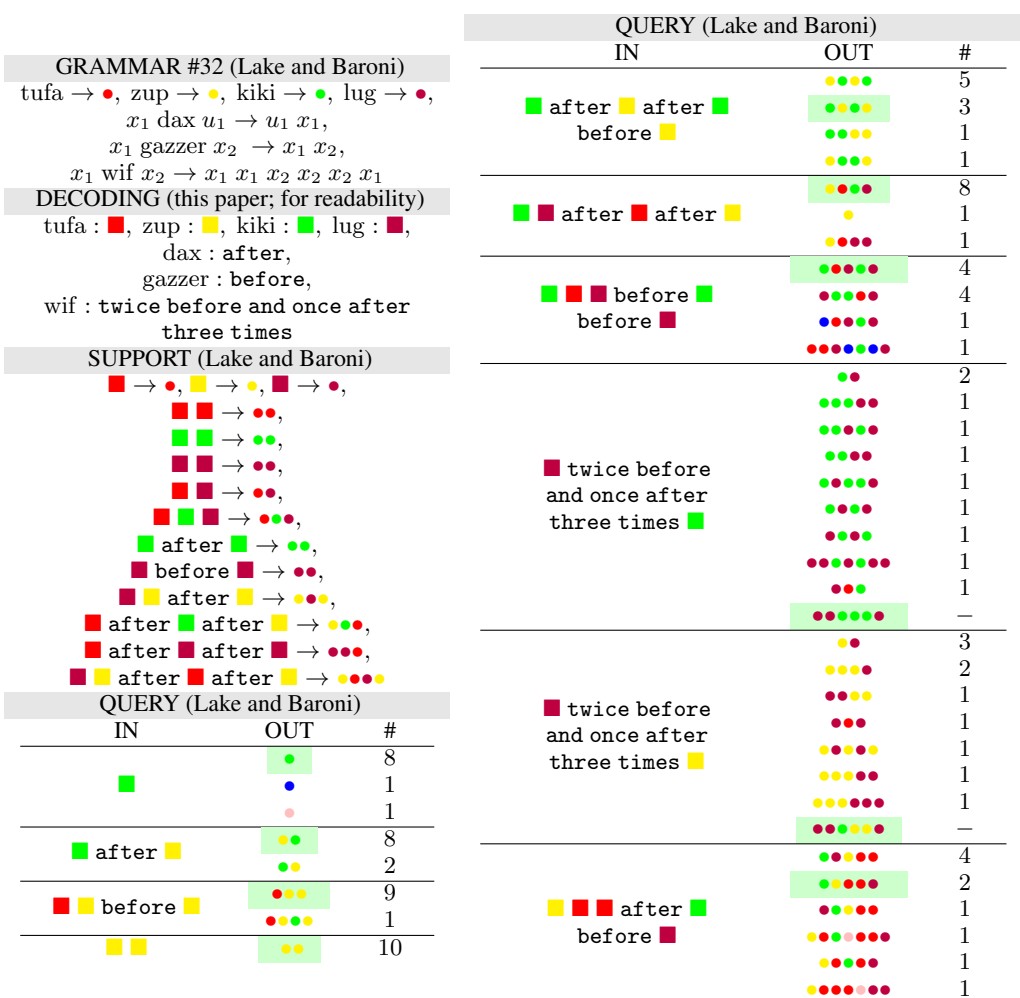

GRAMMAR #32 (Lake and Baroni)

tufa → •, zup → •, kiki → •, lug → •,
$x_1$ dax $u_1$ → $u_1$ $x_1$,
$x_1$ gazzer $x_2$ → $x_1$ $x_2$,
$x_1$ wif $x_2$ → $x_1$ $x_1$ $x_2$ $x_2$ $x_2$ $x_1$

DECODING (this paper; for readability)

tufa : ■, zup : ■, kiki : ■, lug : ■,
dax : after,
gazzer : before,
wif : twice before and once after
three times

SUPPORT (Lake and Baroni)

Table 4: Episode #32 with 10 evaluations for each query example; decoded for better readability. Expected outputs backed with green. (Best viewed in color.)

 **A.4 Complete responses for Lake and Baroni's meta-learning testing-episode #122.**

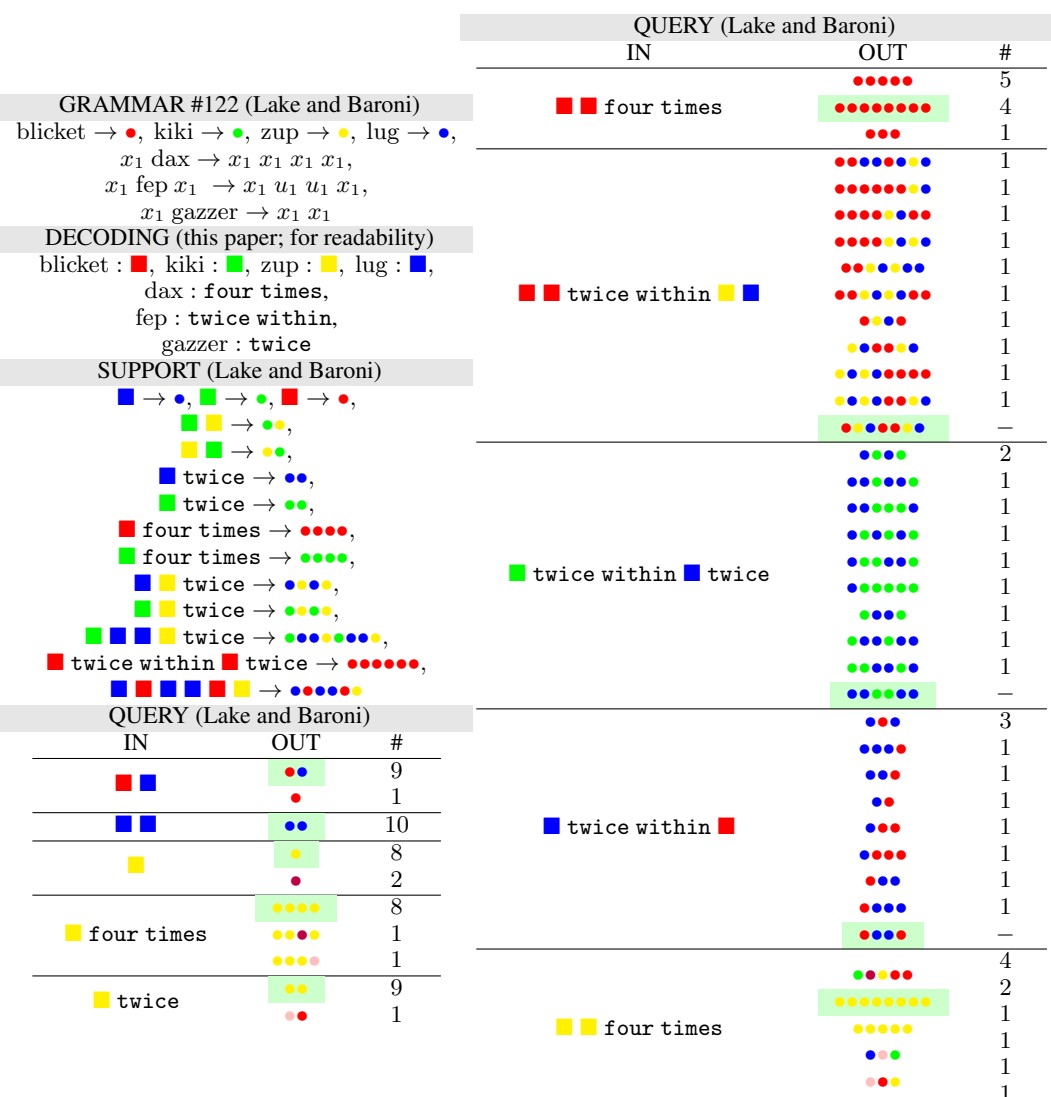

Table 5: Episode #122 with 10 evaluations for each query example; decoded for better readability. Expected outputs backed with green. (Best viewed in color.)

 **A.5 Complete responses for our modified version of Lake and Baroni's testing-episode #1.**

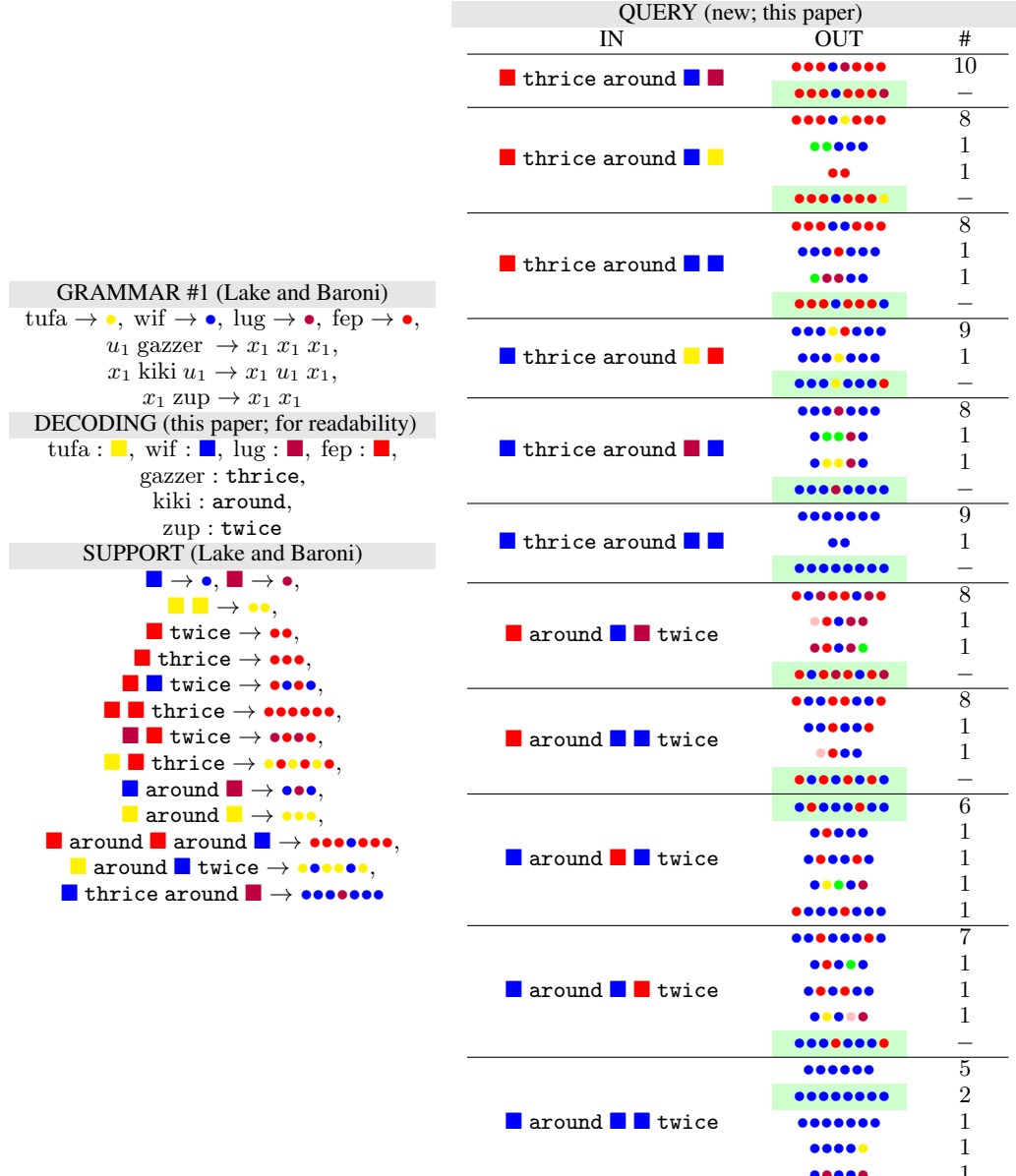

Table 6: Our own query examples for Episode #1 with 10 evaluations each; decoded for better readability. Expected outputs backed with green. (Best viewed in color.)

