# OpenReview forum: "Fodor and Pylyshyn’s Legacy – Still No Human-like Systematic Compositionality in Neural Networks"
_NeurIPS.cc/2025/Position_Paper_Track — Submitted to NeurIPS 2025 Position Paper Track_

### Official Review · Reviewer_wgTd · 2025-07-05

**Significance:** 2
**Presentation:** 2
**Rating:** 3
**Confidence:** 4

**Summary:**

This paper revisits Fodor & Pylyshyn’s 1988 claim that neural nets lack human-like compositionality. It focuses on Lake & Baroni’s 2023 meta-learning seq-to-seq benchmark, rewrites its hidden grammars into a clear pseudo-language, and reruns the released transformer ten times per episode, exposing big accuracy drops and rule inconsistencies.

Key contributions: (i) pinpoints which parts of the Fodor–Pylyshyn critique the benchmark tackles; (ii) empirically shows the model confuses ‘twice’/‘thrice’, mis-parses nested ‘before/around’ strings, and fails beyond length-10 inputs; (iii) sets evaluation criteria—full out-of-distribution ablations or direct inspection of learned codes; (iv) proposes training that merges meta-learning with symbolic memory and iterative self-checking.

The paper’s stance is that the legacy stands: as of 2025 no neural network, meta-learner included, shows human-level compositionality. Real progress, it argues, will come from hybrid models able to store, manipulate and test symbolic rules within neural systems, not from simply scaling one-shot transformers.

**Strengths:**

The paper’s main virtue is that it restates the 1988 Fodor-Pylyshyn challenge and pin-points exactly which parts Lake-Baroni tackle, giving readers a sharp conceptual map before any experiments.

It then faithfully re-runs the public transformer on the same meta-learning episodes, with multiple seeds, and releases full episode-level outputs, so every empirical claim is reproducible and audit-able.

Beyond headline accuracy, the authors recode the hidden grammars into a readable pseudo-language and count unseen rule combinations, revealing that only 179 / 200 validation episodes are novel—a meticulous diagnostic that shows where generalisation really fails.

They further distil two concrete evaluation rules—do exhaustive OOD ablations or directly inspect learned codes—and propose “reflective learners” that iteratively verify and self-correct symbolic hypotheses, pointing to a constructive research agenda.

Each negative result (e.g., confusion of twice/thrice, breakdown beyond length 10, errors on nested before/around) is backed by the published logs, keeping the narrative tight and the reasoning easy to follow even for sceptical readers.

**Weaknesses:**

The study tests only Lake-Baroni’s single seq-to-seq benchmark; adding tasks that vary modality, grammar family, and noise would show whether the critique generalizes.

Results come from ten random seeds without confidence intervals or statistical tests, so readers cannot judge robustness or effect size.

The evaluation caps inputs at 10 tokens and outputs at 8 colors, preventing any look at true productivity or deep nesting limits .

Manual pseudo-language “decoding” risks researcher bias; an automated grammar-induction baseline could verify the claimed mis-parses.

Only the released transformer is re-run; including memory-augmented nets, symbolic decoders, or large instruction-tuned LLMs would ground the comparison.

Focus stays on linguistic rules; alternative views note that compositional meta-learning might manifest differently in vision or motor domains, which the paper sidelines .

Recent work finds emergent systematic skills in scaled LLMs; benchmarking such models could illuminate whether scale, not hybrids, closes the gap.

Clarifying falsifiable hypotheses, releasing code for the reflection loop, and varying support size, OOD length, and architecture would make the argument sharper and more actionable.

**Questions:**

1. You report means over ten random seeds but no confidence intervals or hypothesis tests. Could you provide formal statistics (e.g., bootstrap CIs or paired t-tests) so we can gauge the reliability of the observed accuracy drops?

2. Your pseudo-language “decoding” is manual. How was inter-annotator agreement measured, and could an automated grammar-induction baseline validate the identified mis-parses to dispel concerns about confirmation bias?

3. Inputs are clipped at 10 tokens and outputs at eight color symbols. Have you experimented with longer sequences to reveal whether errors grow gradually or appear abruptly at a specific depth/length threshold?

4. The study reruns only the original transformer. Why not include memory-augmented nets or large instruction-tuned LLMs (which might implicitly build symbolic caches) to test whether scale or architectural tweaks narrow the gap Fodor & Pylyshyn highlight?

5. Many recent works quantify compositionality with information-theoretic or topographic metrics rather than accuracy alone. Would adopting such metrics change your conclusions, and can you share raw model representations to facilitate third-party analysis?

**Alternative Position:**

Yes, and alternative positions are trivial straw-man arguments

**Author Identification:**

No.

**Context:**

2

**Discussion:**

3

**Ethics:**

["NO or VERY MINOR ethics concerns only"]

**Position:**

Yes, the paper argues for or against a position related to machine learning.

**Support:**

2

**Thoroughness:**

5

---

### Official Review · Reviewer_wZ9q · 2025-08-04

**Significance:** 3
**Presentation:** 2
**Rating:** 6
**Confidence:** 4

**Summary:**

This paper argues that neural networks still fail to achieve human-like systematic compositional generalization. To support this view, the authors assess Lake and Baroni’s claim and highlight that their MLC framework lacks key elements necessary for making substantive assertions about systematic generalization. The experimental results reinforce the authors' argument.


**Others**

I reviewed this paper during the ICML track. The authors have made substantial revisions compared to their previous submission, resulting in a clearer and more accessible presentation. Given these improvements, along with the significance of the problem and the value of the contribution, I support the acceptance of the paper this time.

**Strengths:**

1. This paper is well-written overall. The authors clearly outline the history of studies on compositional generalization and recent works, providing a strong background.
2. The authors present substantial evidence demonstrating the limitations of the MLC framework.
3. Understanding systematic compositionality is crucial for identifying the limits of neural network intelligence.

**Weaknesses:**

This paper can be broadly divided into two parts: (1) a discussion on the limitations of neural networks in handling compositionality, and (2) a perspective on meta-learning systems.

However, two main issues arise: (1) the connection between the two parts is rather weak, and (2) while the first part presents clear evidence, the second part lacks strong, convincing arguments to support its claims.

**Questions:**

1) The author claims that "Besides both previous failure modes that are related to incompetence in extracting information from the support examples"(line 208), I am not sure why the author comes to this conclusion.

2) What's the definition of non-systematic error (line 255)

**Alternative Position:**

Yes, and alternative positions are well-considered and addressed by the argument

**Author Identification:**

No.

**Context:**

4

**Discussion:**

3

**Ethics:**

["NO or VERY MINOR ethics concerns only"]

**Position:**

Yes, the paper argues for or against a position related to machine learning.

**Support:**

4

**Thoroughness:**

3

---

### Official Review · Reviewer_gbaB · 2025-08-13

**Significance:** 3
**Presentation:** 2
**Rating:** 6
**Confidence:** 2

**Summary:**

Authors take the view that neural networks have not actually demonstrated human-like compositionality in spite of progress, and in particular, it critiques Lake and Baroni's recent meta-learning framework, suggesting that although they seem to find generalization, this actually reflects memorization of certain patterns in narrow settings. They suggest that evaluations using out‑of‑distribution stress tests and also explicit representations that can be inspected in order to help models avoid non-systematic errors.

**Strengths:**

- Clear reproducible critique about an important perspective from Lake and Baroni
- Broad context across both classical cognitive arguments and modern deep learning
- Very actionable criteria and suggestions for how we can build better meta-learners

**Weaknesses:**

- Very limited set of evidence from a single model and type of benchmark family
- The idea of a non-systematic error is not 100% clear and defined
- Limited evidence that neor-symbolic approaches or RL-guided reasoning could help

**Questions:**

- How would you formally define and quantify a non-systematic error versus a more broadly wrong rule hypothesis?
- To what extent does this criteria apply to other tasks where symbols are less well crystallized and where composition is more fluid?
- What are toy experiments or scaffolds which would demonstrate that neuro-symbolic approaches can be helpful here?

**Alternative Position:**

Yes, and alternative positions are well-considered and addressed by the argument

**Author Identification:**

No.

**Context:**

3

**Discussion:**

3

**Ethics:**

["NO or VERY MINOR ethics concerns only"]

**Position:**

Yes, the paper argues for or against a position related to machine learning.

**Support:**

3

**Thoroughness:**

2

---

### Note · Authors · 2025-08-21

**1-10 Additional Comments:**

We are a bit puzzled by the selection by a reviewer describing our alternative positions as "strawman arguments" without providing any justifications or questions regarding this in the review. Missing such details makes it difficult for us to understand their critique and, consequently, to improve our work.
Maybe this is an issue from the reviewing template (e.g., drop-down menu), in which case we recommend modifying the wording.

Overall, we would have benefited from knowing the details/selection options of the template.

**1-11 Submit Again:**

Probably yes

**1-1 Submission Process:**

3

**1-2 Next Year:**

The timeline and procedure could be communicated more in advance.
Making the review template/rating options public would help with transparency.

**1-3 Future Development:**

See above.

**1-4 Interest:**

["Panel discussions with other position paper authors", "Structured debates on controversial topics"]

**1-5 Thoughtful:**

4

**1-6 Supportive:**

5

**1-7 Technical Aspects Versus Position:**

2

**1-8 Gate Keeping:**

7

**1-9 Camera Ready Changes:**

To reduce confusion (highlighted by reviewer wgTd), we have replaced the term "accuracies" with "success rates" in line 185.

For clarification, we have replaced the introductory sentence in line 208 with the following: "In addition to the previous two failure modes, which are related to the challenges of extracting information from the provided support examples. We thank reviewer wZ9q for pointing this out.

We have added a more detailed explanation of nonsystematic errors. Thus, we now call an error non-systematic when there is no consistent pattern or function in the problem space that can explain its behavior. For the seq2seq transduction example, mistaking "twice" for "thrice" or deriving a wrong rule from ambiguous support can be systematic if the model consistently applies it. We thank reviewer gbaB and wZ9q for pointing out that this may not have been clear in our previous discussion.

**3-1 Review Response1:**

gbaB

**3-2 Reaction To Review1:**

We thank reviewer gbaB for the review and questions and clarify remaining issues below.

(W1) Our position is not a critique of model architecture based on an empirical study. Rather, it outlines **how to argue, test, and train** for systematic generalization and compositionality, emphasizing the importance of structure-sensitive operations and compositional representations for systematic behavior. We thus discourage overstating legitimate, yet gradual, progress toward systematic compositionality, which we observe in recent papers such as the recent LakeBaroni23 framework. Specifically by showing that their model makes nonsystematic errors in their simplified seq2seq environment, we conclude that the model does not in fact exhibit structure-sensitive behavior, which undermines their claim of systematicity and at the same time underlines **our** position on proper testing for compositional skills.

(W2/Q1/Q2) For the seq2seq example, nonsystematic errors can be defined by the absence of functions in the problem space that consistently explain the errors with respect to the structural similarity of the inputs. Mistaking "twice" for "thrice," for example, or deriving a different rule from ambiguous support, could be considered systematic errors if the model consistently applies them. However, a more general definition depends on the structure sensitivity under investigation.

(W3/Q3) We are not claiming to provide evidence. Rather, our goal is to emphasize that these approaches allow for the direct inspection of model representations. This makes validating claims about compositionality easier than with models in which a symbolic level may only emerge during training. Thus, we emphasize the importance of symbolic systems and RL-guided approaches because evaluating systematic compositionality requires assessing the structure-sensitivity of model operations and the compositionality of their representations.

**3-3 Review Response2:**

wZ9q

**3-4 Reaction To Review2:**

We thank reviewer wZ9q for the review and suggestions, and we are glad to hear that the reviewer acknowledges the improvements resulting from the ICML rebuttal. We address the few remaining clarification points below.

(W1) Our perspective on meta-learning systems arose from the idea that evaluating systematic compositionality requires evaluating the structure-sensitivity of model operations and the compositionality of their representations. Although model operations can be evaluated using proper test setups, inspecting model representations necessitates a symbolic level that either emerges from or is built into the model's architecture.

(Q1) With this sentence, we wanted to express that "Failure in rule extraction" and "Non-systematic parsing" show errors related to the challenges of the in-distribution rule extraction task, while "Violating structure-sensitivity" showcases the non-systematicity in OOD queries. For clarification, we will replace the introductory sentence with the following: "In addition to the previous two failure modes, which are related to the challenges of extracting information from the provided support examples."

(Q2) We call an error non-systematic when there is no consistent pattern or function that the model should learn, which can explain its behavior. For example, mistaking "twice" for "thrice" or deriving a different rule from ambiguous support can be systematic if the model consistently applies it. We thank the reviewer for pointing out that this may not have been clear in our previous discussion. We will add this explanation to the section for the camera-ready version.

**3-5 Review Response3:**

wgTd

**3-6 Reaction To Review3:**

We thank reviewer wgTd for the extensive review and suggestions. However, there is an important misunderstanding regarding the scope of our position and the structure of our argument. (W1,W3/Q3,W5/Q4) Namely, our paper is not an empirical case study but rather a position that outlines **how to argue, test, and train for systematic generalization and compositionality**. We hereby discourage overstating legitimate yet gradual progress toward systematic compositionality. The famous LakeBaroni23 framework thus acts as a **demonstrative case** that is sufficient for underlining the relevance of our position on testing for compositional skills, rather than showing that compositionality can not be achieved with related approaches. By demonstrating their model exhibits nonsystematic errors even within their simplified seq2seq environment with limitations on input and output length, we can conclude that they did not achieve structure-sensitive behavior to substantiate their claim of systematicity.

Second, we would like to clarify that we are not arguing that hybrid architectures are necessary to achieve compositionality. Instead, we argue that the symbolic level is crucial for evaluating systematic behavior. Our main point is that structure-sensitive operations and compositional representations are necessary for systematic compositionality. While the former can be evaluated by OOD testing, the latter requires symbolics, at least in post-hoc explanations of internal model mechanisms.

Regarding technical concerns (W2/Q1,W4/Q2): We ran the original seq2seq model with unchanged setup and used post hoc decoding for visualization only. There are no seeds for direct comparison. Variability comes only from individual sampling of the model. The only sensible SD is a measure of the variability of success within an episode (eg, $SD(133)=.43$), using the total counts in App. A.2. We will call it success rate in the camera ready version to reduce the confusion with training accuracy.

---

### Meta-Review · Area_Chair_8Wc9 · 2025-09-12

**Rating:** 3
**Confidence:** 4

**Strengths:**

The reviewers agree on the importance of this topic, situating the discussion within the classical Fodor–Pylyshyn argument, and offering actionable suggestions for future benchmarks.

**Weaknesses:**

However, all reviewers also raise significant concerns, e.g. limited empirical scope, short conceptual precision, and weak methodological robustness.

**Questions:**

see reviewers' questions.

**Ethics:**

no ethical issues

**Thoroughness:**

3

---

### Decision · Program_Chairs · 2025-09-26

Reject